# OpenReview forum: "White-Box Sensitivity Auditing with Steering Vectors"
_TMLR — Accepted by TMLR_

### Review · Reviewer_m3Du · 2026-03-10

**Summary Of Contributions:**

This paper proposes a white-box auditing framework for large language models (LLMs) that evaluates model behavior by intervening in internal representations using activation steering vectors. The central idea is to assess model sensitivity to specific high-level concepts (e.g., gender or race) by perturbing latent representations along directions corresponding to those concepts. The main contributions are:

- The paper introduces a framework that performs concept-level sensitivity analysis inside the model representation space
- The authors propose a sensitivity measure based on the directional derivative of the model output along a concept steering vector, allowing the quantification of how strongly predictions depend on a particular concept.
- Application to bias auditing in decision tasks
- The experiments show that the proposed white-box approach can reveal biases that traditional black-box evaluation methods fail to detect.

**Audience:**

Yes

**Audience Explanation:**

Several communities within TMLR’s audience would likely find this work relevant. It contributes to AI safety and alignment by highlighting limitations of purely black-box auditing. It is also related to interpretability research, as the method builds on representation engineering and activation steering.

**Claims And Evidence:**

Yes

**Claims Explanation:**

The authors present several forms of empirical evidence to support their claims. First, the experiments are conducted across four simulated high-stakes decision settings, which helps demonstrate that the proposed method can be applied in multiple domains. Second, the paper provides a direct comparison with standard black-box auditing approaches. In several cases, the results show that black-box evaluation reports little or no bias, while the proposed white-box method detects significant sensitivity to protected attributes. Third, the authors perform robustness checks by testing alternative perturbation strategies and different steering vectors in order to examine whether the results remain consistent across variations of the method. Finally, the paper includes additional validation experiments, such as name perturbations with varying levels of gender signal strength, to illustrate that the detected sensitivity likely corresponds to meaningful bias risks rather than artifacts introduced by the steering mechanism.


However, several aspects weaken the overall strength of the empirical evidence. First, the experiments rely on synthetic decision tasks rather than real-world deployments, which limits the extent to which the conclusions can be generalized to practical auditing scenarios. Second, the method depends on the assumption that high-level concepts correspond to linear directions in the model’s representation space. Third, the results rely heavily on the quality of the steering vector extraction procedure, and it is not entirely clear how sensitive the auditing outcomes are to factors such as the choice of dataset, the selection of representation layers, or the scaling applied to the steering vectors. Finally, the empirical evaluation is conducted on only a small number of open-weight models, which restricts the ability to assess the generality of the proposed approach across different architectures and training paradigms.

**Requested Changes:**

Several improvements would strengthen the paper. First, the authors should clarify the assumptions behind steering vectors, particularly under what conditions a concept can be represented by a single linear direction in the representation space and what happens when the concept is multi-dimensional or non-linear.

Second, the paper would benefit from a stronger theoretical justification. While the sensitivity metric is derived from directional derivatives, the connection between perturbations in latent representation space and meaningful real-world concept manipulation could be explained more rigorously.

Third, the experimental evaluation could be expanded. Including more model architectures, evaluating larger models, and testing the approach on real decision-support systems or real-world datasets would help demonstrate the broader applicability of the method.

Fourth, a more systematic sensitivity analysis of the steering vector construction would be useful. In particular, the authors could analyze how the results change when using different datasets for steering vector extraction, selecting different layers of the model, or varying the scaling coefficients applied to the steering vectors.

Finally, the interpretation of the reported bias scores should be clarified. The paper should discuss more carefully whether sensitivity to a concept necessarily implies unfair or discriminatory decision behavior, since model sensitivity does not always translate directly into harmful outcomes.

---

### Review · Reviewer_iCbt · 2026-03-22

**Summary Of Contributions:**

This paper proposes a white-box sensitivity auditing framework for LLMs that uses activation steering vectors to manipulate latent concepts (e.g., gender, race) internally and measure model sensitivity via a directional derivative-based metric. The method is applied to four simulated high-stakes decision tasks. The authors find that white-box methods can reveal substantial bias in cases where black-box perturbation-based evaluations show little or even none.

Section 4.5 and 5 is especially strong in revealing the robustness of the white-box methods and the validity of the exposed bias.

Some key weaknesses include:
1. It is unclear whether the proposed methodology is indeed original rather than merely a combination of WMD and Kim et al.
2. Some part of the writing can be improved.

**Audience:**

Yes

**Audience Explanation:**

AI fairness community may be interested.

**Broader Impact Concerns:**

None is needed.

**Claims And Evidence:**

Yes

**Claims Explanation:**

The main claim that white-box steering can reveal biases that black-box methods miss is well-supported by the Credit Scoring and Judicial experiments (Figure 1, Figure 2).

The validity experiments in Section 5 are particularly convincing: Section 5.1 demonstrates through name perturbation that the white-box findings align with real implicit gender signals, and Section 5.2 uses Sobol' index analysis to show that steering isolates the target concept with minimal side effects on other variables.

That being said, the authors, in the Introduction section, claim to have developed "a novel evaluation method that applies steering vectors to manipulate latent concepts within model internals and assess model behavior using a sensitivity metric" but I fail to evaluate whether it is indeed novel since in Section 3, the narrative is such that the method is a straightforward combination of WMD and Kim et al. More clarification is needed.

**Requested Changes:**

1. As discussed above, the authors, in the Introduction section, claim to have developed "a novel evaluation method that applies steering vectors to manipulate latent concepts within model internals and assess model behavior using a sensitivity metric" but I fail to evaluate whether it is indeed novel since in Section 3, the narrative is such that the method is a straightforward combination of WMD and Kim et al. More clarification is needed.

Writing can be improved, more specifically:

2. Section 2.2, while titled as "Black-Box vs. White-Box Evaluations", is basicaly all about black-box models.
3. In Section 2.3, WMD is selected as the method to adapt but it is not even described what it does and why the authors choose to adapt it. More explanation will make the paper more self contained.
4. The opening of Section 3.1 deliver essentially the same message as Section 2 ("black-box methods areinsufficient because they only operate on inputs").
5. The Admissions and Medical tasks don't seem to add a lot to the results. More justification on why they are even considered may be helpful.
6. Section 3.2 is too handwavy.

---

> ### Author Response · Authors · 2026-04-14
>
> **W1: “It is unclear whether the proposed methodology is indeed original rather than merely a combination of WMD and Kim et al.”**
>
> We thank the reviewer for pointing this out. We fully agree that our method builds upon Weighted Mean Difference (WMD) for steering vector extraction and the directional derivatives from Kim et al. for sensitivity measurement. We do not intend to claim any novelty in these methods and will revise the writing to ensure this is clear. Our contribution is adapting these techniques to develop a novel, concrete approach for conducting white-box audits, which the field currently lacks (see the third paragraph of the Introduction).
>
> Prior to our work, WMD and other activation steering techniques were primarily used for mitigation and behavioral control, whereas Kim et al.’s metric was designed for post-hoc interpretability. Neither has been used for systematic evaluation and auditing of LLMs.
>
> We will revise the Introduction and Section 3 in the final manuscript to clarify the novelty of our method and make the distinction between our work and prior work (WMD and Kim et al.) more explicit.
> \
> \
> **W2: The writing**
>
> We appreciate the reviewer’s constructive feedback on the paper writing. We will revise our manuscript accordingly as follows:
> 1. Section 2.2 is titled “Black-Box vs. White-Box Evaluations”, but is mostly about black-box models: We will change the title to “Limitations of Black-Box Evaluations”.
> 2. More explanation on what WMD does and why we adopt it (Section 2.3): We will expand on the mechanism of WMD and the reason for choosing this method; As demonstrated by Cyberey et al, WMD’s steering vectors align more closely with the target concept and provide more precise control over model outputs compared to the prevailing difference-in-means method. This fine-grained control would allow for more accurate sensitivity measurements.
> 3. The opening of Section 3.1 delivers essentially the same message as Section 2: We will remove the repetitive message in Section 3.1.
> 4. Justification for why Admissions and Medical tasks are considered: Even though most models exhibit minimal bias on these two tasks under both black-box and our proposed methods, we decided to include them to demonstrate cases where models pass the audit and to show that our method does not over-report bias with false positives. We will add a brief sentence to Section 4.4 to explain our justification.
> 5. Section 3.2 is too handwavy: We will use a concrete running example (e.g., credit scoring) and map each conceptual step to its corresponding mathematical formulations, with pointers to Sections 3.3 and 3.4.

---

### Review · Reviewer_grEn · 2026-04-22

**Summary Of Contributions:**

The paper aims to address the limitations of black-box evaluation methods of LLM sensitivity to socially-relevant attributes. The paper proposes a white-box sensitivity auditing framework using activation steering. Experiments on bias auditing in four domains (judicial trials, credit scoring, college admissions, and medical diagnosis) show that the white-box method:
1. measures model dependence on attributes that are missed by a black-box method
2. produces results that are more robust against implementation differences
3. makes sensitivity measurements that can be corroborated by input perturbations
4. has little impact on non-protected attributes

**Audience:**

Yes

**Audience Explanation:**

The paper shows that the proposed white-box auditing framework has numerous strengths (principled, shows underlying sensitivity) over a pure black-box approach of "just test various inputs". This is relevant to people who work on LLM interpretability and fairness.

**Broader Impact Concerns:**

If well-received by the community, this paper could impact the way people evaluate and deploy LLMs in socially-sensitive scenarios. Therefore, it is paramount to double check the interpretation of sensitivity as probability, and any ontological claims about revealing actual bias. I elaborate more on both points in the requested changes section.

**Claims And Evidence:**

Yes

**Claims Explanation:**

All claims made in the "Contributions" section are backed up by results.

**Requested Changes:**

Two requested changes:

1. Either make the connection between sensitivity scores and probability claims more rigorous, or remove those claims.

Section 4.4 contains these statements:

> the white-box method indicates that males are 5% more likely to be predicted as having bad credit than females.

and

> However, the white-box method reveals that racially black individuals have close to 20% higher chance of the conviction outcome than white individuals. These results indicate that our white-box auditing method may reveal biases that elude traditional black-box evaluations.

It is not clear how your definition of sensitivity can be interpreted as a probability of a binary outcome (bad/good credit, convict/acquit). This part needs to be rephrased, unless you can explain the connection more rigorously.

2. The use of the word "reveal" can only be justified by resorting to natural language input analysis, which is seemingly contradictory to the position of the paper. Needs to be rephrased.

Various parts of the paper use the word "reveal". This makes it sound like there is some ground truth bias of the model, which is then "revealed" by your method. If that's the case, then you should be able to clearly define the ground truth bias, and compare both the white-box and black-box approaches against it. But that's not the case, because you only have the white and black box approaches, no objective truth. As your section 5.1 shows, you end up resorting to natural language inputs to audit and interpret the validity of your white-box approach anyway. If you want to claim that the white-box approach is better, then it sounds really weird to fall back to the other method to audit your better method.

---

> ### Author Response · Authors · 2026-04-27
>
> We thank the reviewer for their thoughtful feedback and for pointing out the nuances in our writing. We agree with your comment on both points and will make the following revisions to our final manuscript:
> * Sensitivity scores and probability claims: You are correct that we do not have a theoretical justification for showing that our sensitivity scores directly map to a probability of a binary outcome. We will remove the probability claims and revise these statements to strictly reflect the sensitivity scores (e.g., “...the white-box method indicates the model’s prediction for bad credit is more sensitive to males than females, showing a 5% difference in the sensitivity score”).
> * Use of the word “reveal” and ground truth: We agree that using the word “reveal” misleadingly implies our method uncovers the ground-truth bias of the model. Since we do not have a definitive ground truth for model bias, we will revise our language to state that “our method *indicates* potential internal dependencies on the protected attribute, even though the traditional black-box method shows minimal bias”.

---

### Author Response · Authors · 2026-05-15

Dear Reviewers,

We have uploaded a revised manuscript addressing the comments. The main changes include revised language around sensitivity scores and the use of "reveal", clarification of our contributions relative to WMD and Kim et al., and writing revisions as outlined in our response. We are happy to answer any further questions.

---

### Decision · Action_Editor_q9do · 2026-06-03

**Recommendation:** Accept with minor revision

**Audience:**

Yes

**Audience Explanation:**

The work sits squarely within the field of language model evaluation. This is one of the fastest growing fields in machine learning and there are certainly TMLR readers that would be interested in this.

**Claims And Evidence:**

Yes

**Claims Explanation:**

The authors back up each of their claims experimentally:
- model dependence on attributes that are missed by a black-box method
- results that are more robust against implementation differences
- sensitivity measurements that can be corroborated by input perturbations
- little impact on non-protected attributes

They have also agreed to revise the writing of their novelty claim which was contested by reviewers.